

# ProbFire: a probabilistic fire early warning system for Indonesia

Tadas Nikonovas[1], Allan C. Spessa[1], Stefan H. Doerr[1], Gareth D. Clay[2], Symon Mezbahuddin[3]

[1]Department of Geography, Swansea University, Swansea, SA2 8PP, UK
[2]Department of Geography, University of Manchester, Manchester, M13 9Pl, UK
[3]Department of Renewable Resources, University of Alberta, Edmonton, T6G 2E3, Canada

*Correspondence to*: Tadas Nikonovas (tadas.nik@gmail.com)

**Abstract.** Recurrent extreme landscape fire episodes associated with drought events in Indonesia pose severe environmental,
societal and economic threats. The ability to predict severe fire episodes months in advance would enable relevant agencies
and communities more effectively initiate fire preventative measures and mitigate fire impacts. While dynamic seasonal
climate predictions are increasingly skilful at predicting fire-favourable conditions months in advance in Indonesia, there is
little evidence that such information is widely used yet by decision makers.

In this study, we move beyond forecasting fire risk based on drought predictions at seasonal timescales, and (i) develop a
probabilistic early fire warning system for Indonesia (ProbFire) based on multilayer perceptron model using ECMWF
SEAS5 dynamic climate forecasts together with forest cover, peatland extent and active fire datasets that can be operated on
a standard computer, (ii) benchmark the performance of this new system for the 2002-2019 period, and (iii) evaluate the
potential economic benefit such integrated forecasts for Indonesia.

ProbFire's event probability predictions outperformed climatology-only based fire predictions at three to five-month lead
times in south Kalimantan, south Sumatra and south Papua. In central Sumatra, an improvement was observed only at one
month lead time, while in west Kalimantan seasonal predictions did not offer any additional benefit over climatology only-
based predictions. We (i) find that seasonal climate forecasts coupled with the fire probability prediction model confer
substantial benefits to a wide range of stakeholders involved in fire management in Indonesia and (ii) provide a blueprint for
future operational fire warning systems that integrate climate predictions with non-climate features.



## 1 Introduction

Recurrent severe fires in Indonesia release globally significant amounts of greenhouse gases and particulate matter into the atmosphere. Emitted toxic haze can shroud the region for several months (Marlier et al., 2013), with devastating impacts on people's health and livelihoods (Crippa et al., 2016, Lee et al., 2017a), whilst also damaging local ecosystems and biodiversity (Lee et al., 2017b). Every year, during the dry season, fire is widely used for land clearing and in agriculture across the archipelago. In some years, however, anomalously severe droughts do develop, triggering catastrophic

uncontrolled burning events. Two of the biggest such episodes, the 1997-98 and 2015 events each released 0.81 – 2.57 Tg (Page et al., 2002) and 0.21 – 0.53 Tg C (Huijnen et al., 2016, Yin et al., 2016), equivalent to 12 – 40% and 2 – 5% of total global carbon emissions for the year respectively (Boden et al., 2017). Increasingly skilful seasonal climate predictions by dynamic forecasting systems (Doblas-Reyes et al., 2013, Johnson et al., 2019) can potentially be utilised in early warning systems helping to prepare for, and mitigate the worst of the damaging burning events. However, relevant non-climatic

drivers of fire occurrence have to date not been integrated with seasonal climate predictions, leaving an untapped potential for improving early fire event prediction systems. Furthermore,  evaluation of the potential value of such predictions for the decision makers in the region has not yet been carried out to date.

In recent decades, Indonesia's fire problem has been exacerbated by non-climatic drivers such as commodity driven loss and

degradation of primary forests (Turubanova et al., 2018), drainage of peatlands (Hooijer et al., 2012), and conversion of land to industrial plantations and small-holder agriculture (Miettinen et al., 2012). Loss of fire-resilient closed canopy forests (Cochrane et al., 1999, Nikonovas et al., 2020) has resulted in more severe local fire weather due to increased surface heating and substantially warmer microclimates in the deforested landscapes (Sabajo et al., 2017). In peatlands, fire presence was also increased by artificially lowered water table depth due to extensive drainage, which, in combination with increased

surface heating, has exposed more peat to desiccation (Jauhiainen et al., 2014) and the establishment of fire-prone herbaceous vegetation in deforested areas (Hoscilo et al., 2011). These factors, coupled with widespread use of fire by humans for land clearing and crop rotation (Cattau et al., 2016) have dramatically amplified drought sensitivity of fire activity across the region.

The duration and severity of the dry season in different parts of the Indonesian archipelago is modulated by interactions between atmospheric processes associated with inter-annual irregular oscillations in sea surface temperature anomalies in the Pacific and Indian oceans. Drier-than-normal conditions across Indonesia are generally associated with cooler-than-normal sea surface temperatures (SSTs) which occur during strong positive El Niño-Southern Oscillation (ENSO) event (El Niño) and/or positive phase of the Indian Ocean Dipole (IOD) cycle. Reduced precipitation in south Sumatra, south Kalimantan,

and south Papua are typically linked to El Niño events, while dry conditions in north central Sumatra tend to coincide with a positive IOD phase (Aldrian and Susanto 2003, Field and Shen 2008, Field et al., 2016). While the most severe droughts and



widespread burning occurring when both El Niño and IOD are in positive phases (Reid et al., 2012, Pan et al., 2018), short droughts and associated burning events can also develop in neutral ENSO and IOD years, triggered by events such as the dry phase of Madden-Julian Oscillation (Gaveau et al., 2014, Oozeer et al., 2020).


The chaotic nature of atmospheric circulation (Lorenz 1963) prevents reliable numerical weather prediction beyond a couple of weeks (Bauer et al., 2015). Nonetheless, current state-of-the-art dynamic seasonal forecasting systems show skill in seasonal prediction of 2m air temperature and precipitation, especially in tropical regions (Doblas-Reyes et al., 2013, Johnson et al., 2019). The predictability of these chaotic weather parameters at monthly timescales is attributable to

increasingly realistic representation of slowly-evolving SST anomalies associated with the ENSO and IOD variability in seasonal climate forecasting systems (Stockdale et al., 1998, Johnson et al., 2019, Fan et al., 2020, Lavaysse et al., 2020).

Global assessments of seasonal predictability of fire activity have shown that climate information from dynamic models (Turco et al., 2018) and observed sea surface temperature anomalies (Chen et al., 2016, Chen et al., 2020) can be used to

skilfully predict fire occurrence across different regions, including Indonesia. Other studies focused on Indonesia have demonstrated that anomalous drought conditions can be predicted up to a few months in advance (Spessa et al., 2015, Shawki et al., 2017). However, these previous efforts did not integrate non-climate information in fire activity prediction models, and had only aggregated regional resolution.

The climate-fire relationship in Indonesia is strongly regulated at finer spatial scales by human-driven rapid transformation of land cover in Indonesia, particularly in peatland ecosystems (Miettinen et al., 2012, Turubanova et al., 2018, Nikonovas et al., 2020). As such, land cover change and forest fragmentation are critical ingredients for predicting fire activity in Indonesia. No studies have assessed how well the skill of seasonal drought prediction at regional scales translates to fire activity forecasting at fine spatial scales, which would add value to potential users of the forecasts such as fire managers,

forest conservation and peatland protection agencies. While the integration of non-climate information datasets, development of high spatial resolution probabilistic forecasting models and long-term performance validation have been identified by the previous studies as key requirements for building future early warning systems and increasing the usability of the seasonal climate information in fire management (Spessa et al., 2015, Chen et al., 2016, Turco et al., 2018), these challenges have not yet been addressed.


This study aims to (i) move beyond seasonal forecasting of fire activity solely as a function of climate variables, (ii) provide a blueprint for future operational landscape-scale fire forecasting systems and (iii) evaluate the system from a potential user's perspective in terms of skill and economic utility. Specifically, we developed a probabilistic early fire warning system (ProbFire) for Indonesia that integrates information from ECMWF SEAS5 seasonal climate forecasts and non-climate

datasets and produces probabilistic fire event predictions at 0.25 degree spatial resolution with monthly time steps. ProbFire





performance was assessed using MODIS active fire observations during the 2002-2019 period. To assess the added value of SEAS5 seasonal forecasts, ERA5 climatology-based model was used as a benchmark. In addition to evaluating model skill at predicting observed fire occurrences, we also assessed the economic value and benefits of ProbFire predictions for potential stakeholders in Indonesia and beyond.

**2 Data**

**2.1 Fire activity data**

As a proxy of fire activity across Indonesia this study used the Collection 6.1 active fire dataset MY(O)D14 (Giglio et al., 2016) based on Moderate Resolution Imaging Spectroradiometer (MODIS) imagery at thermal wavelengths. The product contains centre coordinates of MODIS pixels (~1 km at nadir, ~10 km at the extreme sensor view edge) which were flagged 110 as active fires by the thermal anomalies algorithm.

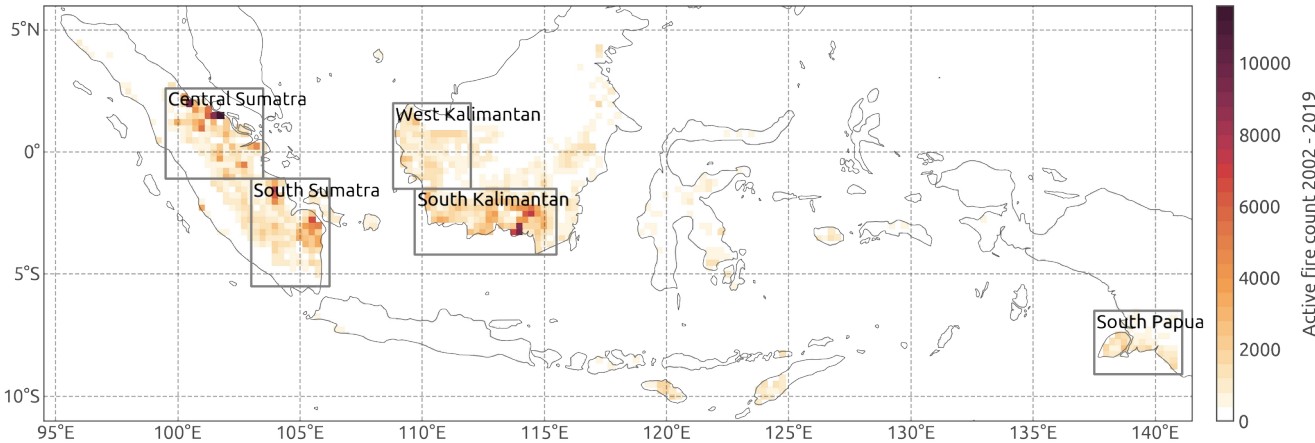

**Figure 1: Study region showing total MODIS active fire counts for the 2002-2019 period at 0.25° resolution. Also shown are the bounding boxes of the sub-regions used in the study.**

The MY(O)D14 product has been shown to perform well in detecting large fire events and to have low false alarm rate. 115 Validation of the product for Equatorial Asia region indicated 8% error of commission (Giglio et al., 2016). Low levels of false detections were also confirmed during the 2013 burning episode in north Riau, where 96% of MODIS active fire pixels were found to fall within the burned area extent estimated using higher resolution imagery (Gaveau et al., 2014). Omission errors for small fires of the MODIS active fire product are inevitably high due to the relatively coarse spatial resolution of the sensor (pixel size 500m at nadir). However, for fires over 0.125 km2 in size estimated omission error was 10%, while for 120 fires of 0.250 km2 or larger, omission error was below 5% (Giglio et al., 2016). Although low temperature smouldering peatland fires are generally more difficult to detect using thermal anomalies algorithms (Giglio et al., 2016), however, such





fires typically have long residence times and as a result detection probability increases with each satellite overpass. A recent study comparing fire emission inventories based on MODIS burned area and active fire datasets for Indonesia showed that active fire-based emission models reproduce regional aerosol optical thickness more accurately when compared to area

burned methods, resulting in a smaller underestimation of fire activity in extreme burning years (Liu et al., 2020).

### 2.1.1 Fire occurrence patterns in Indonesia and prediction objective

The MODIS active fire detections were aggregated to 0.25° spatial grid and monthly time step. Monthly active fire counts were used both as model training targets and for prediction validation. Most of the grid cells (~80%) did not have any active fires for the given month. The counts for the grid cells with active fire pixels were highly skewed, with the majority

containing very few fire detections, while the relatively low number of grid cells with detected fire presence had very high monthly counts (up to ~2500). Approximately three quarters of the grid cells with active fire detections had 1-10 fire pixels, while the remaining upper quartile (~5% of the total dataset) had > 10 active fire detections per month. Importantly, while the number of grid cells having low active fire pixel count (1-10) show a clear pattern of Indonesia's dominant dry season (Aldrian and Susanto 2003), there are only small differences when comparing the fire grid cell counts for different years

(Fig. 2a). In contrast, the number of the top quartile of all fire-containing grid cells varied considerably more between years. Total active fire counts depicted in Fig. 2b exhibit even greater interannual variability indicating that most of the region's fire impacts can be attributed to the grid cells containing > 10 fire pixels.

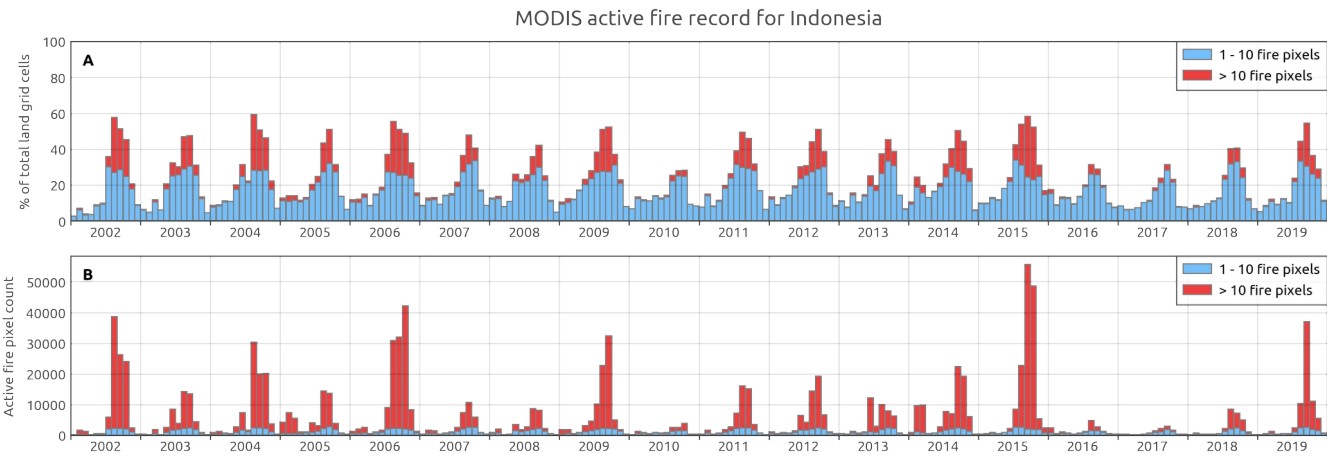

**Figure 2: MODIS active fire record for Indonesia during 2002-2019 period. a) % of total land grid cells in the study region (n =**
**2080) with active fire detections. The two categories shown are for grid cells with relatively low fire counts (1-10 fire pixels per month) and higher degree of fire presence (> 10 fire pixels) in blue and red colours, respectively. b) Total monthly MODIS active fire counts attributable to grid cells of the two categories.**



The main objective of ProbFire is to predict the probability that the monthly active fire count will exceed a given threshold. While we present results of predicting both monthly active fires count > 0 and monthly active fire count > 10 cases, our
analysis focuses on predicting the latter class because it's higher impacts and importance to fire management.

## 2.2 Climate variables

This study used three climatic variables as inputs for prediction of fire activity in Indonesia: total precipitation, air temperature and relative humidity. These climate indicators are strongly linked to fire occurrence and as a result are widely used as key inputs for calculating various fire danger indices (van Wagner and Forest 1987, Dowdy et al., 2009). The
variables were obtained from the European Centre for Medium-Range Weather Forecasts (ECMWF) gridded reanalysis and long-range forecasts products distributed via the Copernicus Climate Change Service. For model training, validation and for computing climatological values we employed the ECMWF's ERA5 reanalysis dataset, while for predictions of fire occurrence probability at one to six months lead times we have used ECMWF's SEAS5 long-range forecasting model simulations.


ERA5 is the latest version of ECMWF reanalysis products. It is based on the Centre's Integrated Forecast Systems coupled atmosphere-ocean model simulations constrained with many assimilated satellite-based and in-situ observational datasets (Hersbach et al., 2020). The ERA5 product used in this study has a regular longitude/latitude grid with 0.25° spatial resolution and 1-hourly time step. We have resampled the ERA5 weather parameters to monthly values by computing
monthly mean 2m temperature, mean monthly 2m relative humidity and total monthly precipitation.

SEAS5 is the fifth generation ECMWF's seasonal forecasting system and has been operational since 2017 (Johnson et al., 2019). The system consists of 51 ensemble members which are initialized on the first day of every month and simulate the state of the atmosphere for a seven-month period. The individual ensemble members are initialized using perturbed initial
conditions and atmospheric model parameters to represent uncertainties associated with the initial state and missing or misrepresented model processes. While the system consists of 51 ensemble members when operated in forecasting mode (since 2017), for the years prior to 2017 SEAS5 system was initialized using only 25 members producing climate hindcasts (alternatively termed reforecasts) for the period covering 1981-2016. In this study, we used the same subset of 25 SEAS5 members which were available for the whole of the study period covering 2002 through 2019, and we also used term
forecasts in describing SEAS5 data from both hindcast and forecast periods. The spatial resolution of the SEAS5 product was one-degree, while the temporal step was one month.

While mean 2m temperature was readily available and total monthly precipitation was simply calculated from precipitation rates given in the respective ERA5 and SEAS5 products, relative humidity was derived from 2m temperature and 2m dew
point temperature using August-Roche-Magnus approximation (Alduchov and Eskridge 1996):




$$Rh = 100\, \frac{\exp\!\left(\dfrac{17.625\,td}{243.04+td}\right)}{\exp\!\left(\dfrac{17.625\,t}{243.04+t}\right)} \qquad (1)$$

where *Rh* is relative humidity, *td* is 2m dew point temperature and *t* is 2m temperature. In total, we used eight climate features as inputs into ProbFire: total monthly precipitation, total monthly precipitation for the five preceding months (t-1 to t-5), mean monthly temperature and mean monthly relative humidity. Precipitation for the months preceding the month of interest was included to characterise long term build-up of drought conditions.

### 2.2.1 SEAS5 bias and variance adjustment

Raw SEAS5 model ensemble forecasts, like any other long-range climate modelling system outputs, are affected by systematic model biases and drift and as a result, forecast climatology (for example long-term mean and variance) is often significantly different from the observed climatology (Doblas-Reyes et al., 2013, Johnson et al., 2019). Furthermore, publicly available SEAS5 forecasts have a spatial resolution of one degree and consequently cannot represent local conditions well, particularly in coastal and mountainous areas (Fig. 3).

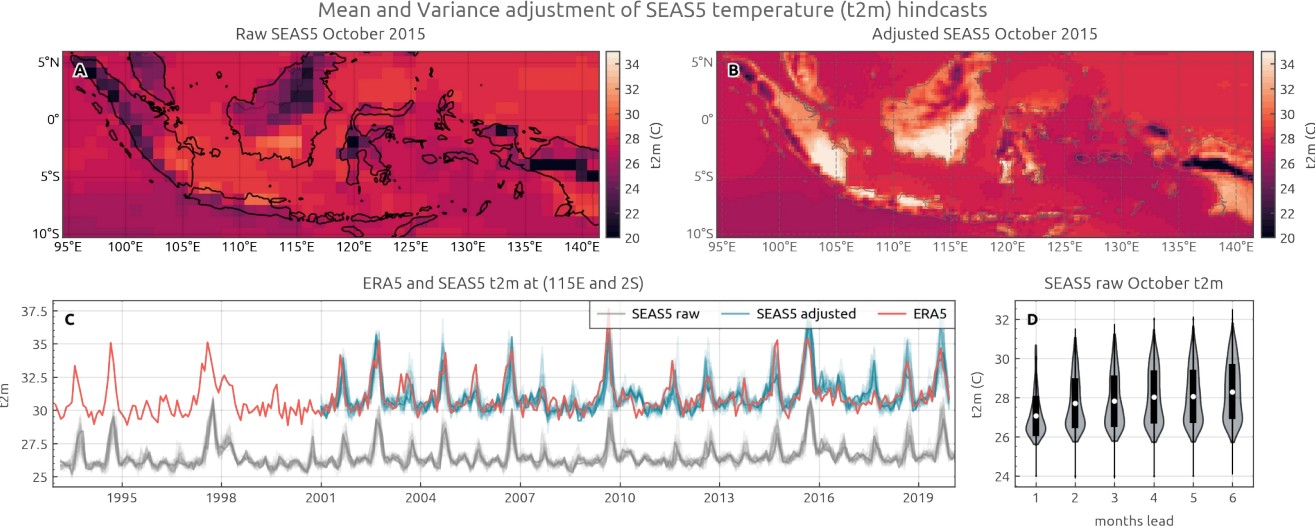

**Figure 3. Illustration of mean and variance adjustment applied to raw SEAS5 ensemble member forecasts. A) Raw SEAS5 member = 1 temperature at 2 metres (t2m) for October 2015 with lead time = 3 months. B) October 2015 mean and variance adjusted t2m of the same SEAS5 member based on calibration against overlapping 1993-2019 series between ERA5 reanalysis and SEAS5 forecasts. C) SEAS5 ensemble member raw and bias adjusted t2m and ERA5 t2m at 115E, 2S location for the 1993-2019 period (bias corrected SEAS5 is only shown for the study period 2002-2019). D) Mean SEAS5 member t2m for all October months in the record for different lead times, showing ensemble mean drift (warming in this case) and increasing spread.**





To address SEAS5 biases and to downscale of SEAS5 hindcasts to match the spatial resolution of ERA5 reanalysis (0.25

degree), we performed a mean and variance adjustment (MVA) of the raw SEAS5 outputs. The method has been extensively

applied in seasonal forecasting (Barnston, 1994, Doblas-Reyes et al., 2005) and has been shown to have similar performance

when compared to more complex and computationally intensive methods (Manzanas et al., 2019).

The MVA was applied in two steps. First, the raw SEAS5 forecast data at one-degree resolution were re-gridded to match

the 0.25 degree resolution of the ERA5 reanalysis data using nearest-neighbour interpolation. Second, the mean and variance

of monthly hindcasts for each SEAS5 ensemble member were transformed to match those ERA5 values of the 1993-2019

period for that grid cell following Eq. (2):

$$fcor_{m,t} = \left(fraw_{m,t} - \underline{y}_t\right)\frac{\sigma_o}{\sigma_f} + \underline{o}_t \tag{2},$$

where $fcor_{m,t}$ and $fraw_{m,t}$ are the mean and variance adjusted and raw SEAS5 hindcast ensemble member $m$ for month $t$,

$\underline{y}_t$ is SEAS5 ensemble mean of all times t, $\underline{o}_t$ is ERA5 mean for the month $t$, $\sigma_o$ denotes ERA5 standard deviation and $\sigma_f$ is

SEAS5 ensemble standard deviation for the calibration period (Fig. 3). The calibration period was 1993–2019, as determined

by the availability of both SEAS5 hindcasts and ERA5 reanalysis on the Copernicus Climate Change Service system

(https://cds.climate.copernicus.eu).

**2.3 Non-climate variables**

In addition to the climate variables, we used environmental features which are closely associated with fire occurrence in the

region. These datasets were: per-grid cell peatland extent, past fire activity, primary forest cover extent, primary forest loss

in previous year and secondary forest loss in previous year (described in detail in the following paragraphs). In contrast to

climate features which had a monthly time step, all the environmental features had annual time resolution except for peatland

extent which was fixed for the whole study period. While numerous other sources of potential feature data exist, they were

not selected because they did not cover the whole of Indonesia for the full study period, and/or did not have at least an

annual time step. This last criterion was particularly important for determining selection of datasets because the fire

prediction model was built to form the basis for an operational early fire warning system.

**2.3.1 Past fire activity**

In many parts of the region, in deforested and unmanaged peatlands in particular, the fire return interval is short due to

frequent repeated anthropogenic burning (Cattau et al., 2016). Frequent fires prevent forest regrowth, and the landscape

becomes dominated by flammable grasses (Hoscilo et al., 2011). The positive feedback between fire and vegetation means

that areas which did experience burning in the past are more likely to burn again. To represent fire occurrence in previous





years as a model input, the maximum monthly active fire detection count for each 0.25° grid cell in the years preceding the prediction year was used.

We used two different active fire products to cover past fire observations for all years in the study period (2002-2019). MODIS active fire record was extended beyond 2002 with Along Track Scanning Radiometer (ATSR) World Fire Atlas
(WFA) night-time fire monthly counts for the 1997–2001 period. This was done to reduce the impact of the lack of past fire observations for first few years in the study period on the model performance. The ATSR WFA night-time fire product contains several times fewer active fire detections when compared to the MODIS product due to lack of day-time retrievals (Arino et al., 2012), and as a result, pre-2002 maximum monthly counts are underestimated when compared the MODIS estimates. However, this step was important to identify areas affected by the 1997-1998 El Niño event and the associated fire
episode which was one of most severe in Indonesia's history (Page et al., 2002)

### 2.3.2 Forest cover features

Loss and degradation of primary forest cover in recent decades has been closely associated with an increase in fire occurrence in the region (Langner et al 2009, Field et al., 2016). Undisturbed humid primary forests in Indonesia are extremely fire-resilient (Cochrane et al., 1999, Nikonovas et al., 2020) and can resist ignition even during prolonged
droughts. By contrast, industrial plantations and agricultural land, which are replacing primary forests, have substantially higher fire activity rates (Nikonovas et al., 2020). We used two high resolution Landsat data-based tree cover datasets used to represent changes in forest cover during the study period at 0.25° spatial resolution and at an annual time step. A co-located analysis of primary forest cover extent in the year 2000 (Morgano et al., 2014) and version 1.6 of the global annual forest cover loss dataset (Hansen et al., 2013), which covers 2001 to 2018, was performed to determine annual primary forest
cover fraction, primary forest loss and secondary forest loss. Pixels classed as primary forest in the year 2000 were matched with the annual forest loss pixels for years 2001-2018. Firstly, we determined if the estimated forest loss had occurred in primary or secondary forest areas. Secondly, primary forest cover loss for each year was derived by subtracting cumulative primary forest loss from the year 2000 primary forest extent estimate. Following these two steps, the annual primary forest cover, primary forest loss and secondary forest cover loss estimates at Landsat pixel level were aggregated to the study's
0.25° resolution.

Definitions of forest cover and primary forests in this study follow the definitions given in the global forest cover loss and primary forest extent of the year 2000 products. Both datasets considered all Landsat pixels with tree height of > 5 m and canopy cover of > 30% as forest cover. Primary forest was defined as a contiguous block of > 5 ha of natural forest which
has not been cleared in recent decades. Note that the definition of primary forest includes both intact and degraded primary forest types (Morgano et al., 2014). Forest cover loss in the annual forest cover loss dataset was defined as a stand replacement disturbance. Both Landsat-based forest cover datasets were found to agree well with alternative estimates. The

primary forest extent of the year 2000 dataset showed approximately 90% agreement when compared to the primary forest map of the year 2000 issued by the Ministry of Forestry of Indonesia (Morgano et al., 2014), while validation of tree cover
loss suggested that forest loss was correctly flagged in more than 80% of the cases (producer's accuracy 83.1%) in tropical regions (Hansen et al., 2013).

### 2.3.3 Peatland fraction

Deforestation and drainage of the region's carbon-rich peatlands in recent decades has rendered large amounts of near-surface peat vulnerable to frequent repeated burning (Hoscilo et al., 2011). Intentional fires in peatlands that are ignited to
clear land and prevent vegetation regrowth often develop into uncontrolled sub-surface peat combustion events which may last weeks or even months (Widyastuti et al., 2020). As a result, the region's peatlands experience fire occurrence rates up to several times higher when compared to non-peatlands (Vetrita and Cochrane 2020, Nikonovas et al., 2020). To represent elevated fire activity in peatland areas we estimated peatland fraction in the 0.25° grid cells using the high-resolution vector maps of peatland distribution and carbon content by Wahyunto and Suparto (2004). The vector maps were rasterized to 0.01°
grid. Any cells whose centroid was inside the peatlands polygons were considered as peat areas. Following the rasterization step, peatland's fraction at 0.25° resolution was determined from the number of 0.01° cells classed as peatlands falling within the lower resolution cells.

### 2.3.4 Sub-region identifier features

Drivers of fire activity vary across different parts of the archipelago have different fire activity rates even when experiencing
comparable drought conditions (Aldrian and Susanto 2003, Field and Shen 2008, Field et al., 2016). To enable the model to represent regional differences in drought sensitivity across Indonesia, we used additional five features representing binary identifiers for each of five sub-regions within Indonesia (Fig. 1).

### 3 Model description and experimental setup

### 3.1 The model

To predict fire occurrence probability we used a multilayer perceptron (MLP) classifier (i.e. a shallow artificial neural network) (Hastie et al., 2009). The main reason for choosing a MLP model was the fact that MLP's do produce well-calibrated probabilities, while at the same time being able to approximate more complex non-linear relationships when compared to simpler probabilistic prediction models such as logistic regression (Niculescu-Mizil and Caruana 2005, Guo et al., 2017).






The model consisted of three fully connected layers; a layer with 18 inputs (see fig. S3), one hidden layer with 15 nodes and an output layer with two nodes. For the hidden layer rectified linear unit (ReLU) activations were used, while sigmoid activation was applied to the output layer nodes to obtain class (active fire counts below or above the threshold) probabilities. The model weights were optimized employing LBFSG solver with learning rate value of 0.001 and cross entropy loss function using L2 regularization alpha parameter value of 1. The input features (climate parameters and land cover information) were scaled to zero mean and unit variance. The model architecture and parameter setup were determined empirically based on the model's performance on validation data. The model (github.com/ToFEWSI/ProbFire) was implemented in the Python 3 programming language using the scikit-learn machine learning library (Pedregosa et al., 2011).

### 3.2 Model validation design

To evaluate ProbFire performance, we employed a leave-one-year out training and validation dataset splitting strategy. This approach provides a more realistic representation of the potential of the model to forecast fire occurrence probabilities for future fire seasons. The whole 17-year record was used (2002-2019), and the MLP model was iteratively trained using 16 years worth of ERA5 reanalysis climate and land cover data, and predicting fire probabilities for the left-out year. For example, fire occurrence probabilities for 2006 were predicted and evaluated using data from all years except 2006 for model training. This resulted in 17 different realizations of the model all having different weights and biases, due to different subsets of the dataset being used for training.

### 3.3 ERA5-based predictions

The first set of model predictions was made using ERA5 reanalysis monthly climate values employing the leave-one-year out strategy. This set of predictions represents the base model and the best-case scenario of this study's fire activity prediction results.

### 3.4 SEAS5-based predictions

ProbFire prediction of fire probability at lead times of 1-6 months was based on SEAS5 climate forecasts for the corresponding lead times. Total precipitation for the previous months (t-1 through t-5) was also based on SEAS5 values for the months within the lead time window, while ERA5 precipitation for the previous months was used if those months were outside the lead time period. For example, prediction for October 2015 at 3 months lead time was based on SEAS5 hindcast issued in August 2015. Precipitation for the preceding months t-1 and t-2 was also based on the SEAS5 hindcasts issued in August, meanwhile total precipitation for the months t-3 through t-5 was derived from ERA5 precipitation rates for July-May 2015. This approach enabled us to utilize all the observational information available at the time when forecasts were issued.





### 3.5 Climatology model

Potential skill and value of long-range fire predictions based on SEAS5 seasonal climate forecasts was benchmarked against climatology-based model predictions. The climatology model had the same input features, except that SEAS5 forecasts were substituted with ERA5 mean values for the 1993-2019 period for a given month. Like the forecasting feature setup, climatological values of total precipitation for the preceding months were used for the months within the forecasting time window, otherwise ERA5 total precipitation was used. For example, climatology-based prediction for October 2015 at lead time 3 months was constructed using mean climate values for October 1993-2019 and climatological values of total precipitation for September and August (t-1 and t-2), but ERA5-based values were used for total precipitation at months t-3 to t-5.

### 3.6 Model performance evaluation

### 3.6.1 Skill scores

To assess model performance, we used reliability diagrams (Murphey et al., 1992), probability of detection and false alarm rate analysis (receiver operating characteristic) (Mason 1982) and the Brier score (Murphy 1973). Reliability diagrams inform how well predicted event occurrence probabilities correspond to the actual observed event frequency. For example, we would have a reliable forecast if taking all cases when 70% event probability was issued, the event would have occurred in close to 70% of those cases. The reliability diagrams were calculated by splitting predicted probabilities into 10 equally spaced bins in a range of [0, 1] and with a step of 0.1, and determining fire event occurrence frequency for each of the bins. To compliment the reliability diagrams, we also constructed prediction histograms, which indicate forecast sharpness. Sharpness is a measure of the ability of a forecast to issue a range of probabilities. It is a desirable property of a forecasting system, because forecasts that issue low or high event probabilities are potentially more useful. In contrast, while a forecast that often gives probabilities close to event climatological frequency may be reliable, it lacks sharpness and hence is of little use for decision-makers.

The probability of detection expresses the fraction of all events that were correctly classified, while false alarm rate indicates the fraction of predicted events which did not occur:

$$pod = \frac{hits}{(hits + misses)} \tag{3}$$

and

$$far = \frac{falsealarms}{(hits + falsealarms)} \tag{4}$$

here *pod* refers to probability of detection, *far* refers to false alarm rate, *hits* equals the number of events that have been correctly classified as events, *misses* is the number of events that were not predicted and *false alarms* is the number of




predicted events which did not occur. The probability of detection is sensitive to hits, but ignores false alarms, while false

detection rate is sensitive to false alarms but ignores misses. Both scores may be artificially inflated, by increasing and

reducing the number of event forecasts in the case of probability of detection and false alarm rate, respectively. While both

scores can indicate if the forecasts are potentially useful, they are calculated at a particular probability threshold. In reality,

different users might benefit from choosing different probability thresholds at which they decide to act. Receiver operating

characteristic (ROC) curve addresses this by showing both probability of detection and false alarm rate at a range of

increasing probability thresholds. The metric indicates the ability of the forecasting system to discriminate between events

and non-events. The area under the receiver characteristic curve is a single number summary score which is used in this

study to compare receiver characteristic curves obtained by different models.

The Brier score is a metric that is widely used to evaluate probabilistic predictions (Murphy 1973, Gneiting and Raftery

2004). Conceptually it is similar to mean squared error, but rather than measuring difference between observed and predicted

real values, Brier score evaluates difference between predicted probability in the range [0, 1] and event occurrence:

$$Brier score = \frac{1}{n} \sum_{t=1}^{n} \left( f_t - o_t \right)^2 \qquad (6)$$

where $f_t$ is the probability of $t^{th}$ forecast, and $o_t$ is 0 if the event didn't occur and 1 if it did. The score takes values between

0 and 1, with smaller values indicating better skill. The Brier score is sensitive both to discrimination and calibration

(reliability), and it is strictly a 'proper' score. The latter property forces forecasters to issue a probability which is equal to

their true expectation (Gneiting and Raftery 2004). In contrast to proper scores, 'improper' scores can be improved by

'hedging', i.e. issuing probabilities which are different from the true expected probability. The Brier score is sensitive to

class prevalence and suffers from becoming vanishingly small for extremely rare events. As a result, it only makes sense to

compare the scores of different forecasts for the same sample.

**3.6.2 Relative value of the forecasts**

The scores discussed above are useful in assessing forecast skill and for comparing performance of different models,

however, they do not explicitly reveal if the decision-makers would benefit from using the proposed forecasting system.

Indeed, it is possible for forecasts to be simultaneously skilful but not useful. The cost-loss ratio analysis (Murphy 1985,

Richardson 2000) addresses the usefulness question directly by providing an assessment of potential economic value of the

forecasts. This model, while simplistic and of limited applicability when accounting for non-economic impacts, nonetheless

allows us to quantify the value of forecasts for a range of users with a range of specific cost-loss ratios.



For example, if the event is a 'peatland fire', and the action is 'fire preventative measures', then loss would equal the total economic loss caused by the fire event and cost would be the total economic cost of the preventative measures. Each time a decision maker takes no action and fire event occurs, it results in a loss. Alternatively, every time the decision maker acts it incurs a specific cost. Every time action is taken and the predicted fire event occurs, the difference between the reduced loss and invested costs constitutes the total amount saved. A reliable forecasting system can inform the decision maker when (and where) to act to minimize total expenditure. As a result, such a forecasting system has a potential economic benefit, and the cost-loss analysis indicates this potential economic gain, or in other words, the relative value of the forecasts.

This relative value is expressed as a fraction of value of a perfect (theoretical) forecast and indicates improvement over a scenario when the only information available to the user is climatological event occurrence frequency. The relative value of a forecast depends on the user-specific cost and loss and is positive over a limited range of cost-loss ratios. If cost is larger than loss, not acting is always better, and vice-versa, if cost is very low in relation to the potential loss, always acting is the best option. Both these scenarios negate the need for a forecasting system. The potential value of the forecasts is highest at the cost-loss ratio value which is equal to the event climatological frequency. Benefits vary for different users with different cost-loss ratios, and, assuming reliable probabilistic forecasts, an optimum probability decision threshold is equal to user cost-loss ratio (Richardson 2000). As a result, users with high cost-loss ratios would benefit most from choosing to act at higher event probability thresholds and vice versa. In this study, relative value was calculated for a range of cost-loss ratios [0.001, 1] following the Eqs. (7) and (8):

$$relative\ value = \frac{\frac{c}{l}(hits + false\ alarms - 1) + misses}{\frac{c}{l}(Pclim - 1)} \quad if\ \frac{c}{l} < Pclim; \tag{7}$$

$$Relative\ value = \frac{\frac{c}{l}(hits + false\ alarms) + misses - Pclim}{\frac{c}{l}(Pclim - 1)} \quad if\ \frac{c}{l} \geq Pclim; \tag{8}$$

where $\frac{c}{l}$ is cost to loss ratio and $Pclim$ is the climatological probability of occurrence of the fire event (i.e. active fires > 10) for the sub-region of interest. Note that relative value (same as probability of detection and false alarm rate), is calculated at a particular probability threshold, in effect transforming the continuous probabilistic forecasts to binary fire event vs no fire event predictions to derive hits, false alarms and misses. As a result, relative values are derived for a range of probability thresholds indicating potential benefits for users with different cost-loss ratios.





### 3.6.3 Mean SEAS5 ensemble probability

In contrast to the traditional ensemble evaluation methods that derive probabilistic forecasts from distribution of deterministic predictions of the individual ensemble members, the modelling method employed by this study predicts probabilities of fire counts exceeding a given threshold for all 25 members of SEAS5 ensemble. For deriving ensemble mean skill scores we combined the estimates based in individual members into a single probability estimate by computing simple equally weighted average probability.

**4 Results and discussion**

### 4.1 Reliability of probability prediction

ProbFire forecasts of active fire counts > 0 in any given grid cell in any particular month generally exhibited good degree of reliability (Fig. 4). Reliability diagrams of ERA5-based prediction for the study years indicate, with a few exceptions, that for most years reliability curves were close to the perfect diagonal line (Fig 4a). For low fire activity years, the predictions

were generally overconfident (i.e. probabilities higher than the observed fire event frequency). Predictions were less reliable only for two of the relatively high fire activity years, 2002 and 2019. Predictions for 2002 were underconfident, meaning that the model generally underestimated fire event occurrence frequency for that year. Active fires were more frequently detected in grid cells for which the model issued low probabilities. This underestimation may be because 2002 was the first year in the record which had no prior MODIS active fire observations and also only a limited primary forest loss record. Although

we tried to extend back the MODIS observations beyond 2002 with the ATSR WFA night-time active fire dataset, the later product has much lower fire counts and could not provide sufficient record of fire activity prior to 2002.




**Figure 4: Reliability diagrams of active fire counts > 0 case occurrence probability predictions. Inset axes show ERA5-based probability prediction frequency histograms. a) ERA5-based prediction for all of Indonesia reliability curves for each year in the record. The colour of the lines corresponds to active fire > 0 cases count for the year. (b-f) Mean reliability curves for the sub-regions (Fig. 1), showing ERA5 (red) and SEAS5-based ensemble mean prediction reliability curves at different lead times (shades of grey, bottom legend). Dotted lines indicate perfect reliability (1:1 fit).**

By contrast, predictions issued for the year 2019 were too high across the whole range of probabilities. This overestimation could be due to several factors. Firstly, the 2019 drought was driven by positive IOD, while ENSO was neutral. Secondly, since the 2015 burning episode, the Indonesian government has implemented further policies aimed at reducing deforestation and fire occurrence (Hergoualc'h et al., 2018, Carmenta et al., 2020), which may have contributed to lower than expected fire detections in 2019.

ERA5 and SEAS5-based prediction probabilities for active fires > 0 pooled for all years, but split into different sub-regions, generally indicate good reliability (Fig. 1b-f). However, there are some notable differences when comparing the spatial domains. Notably, all predictions for south Kalimantan and south Papua indicate overconfidence, while forecast (SEAS5-



based) probabilities for west Kalimantan were underconfident. SEAS5-based predictions performed o generally well for all
regions and all lead times, exhibiting only a gradual degradation in reliability of high probability predictions with increasing

lead time. There were noteworthy differences when comparing the ERA5-based probability histograms for different sub-
regions. Predictions for central Sumatra and west Kalimantan lack sharpness, a property which is manifested by a relatively
low number of very high probabilities issued for those regions. In contrast, the model was able to discriminate between no
fire and active fire count > 0 cases more easily in south Sumatra, south Kalimantan and south Papua. This difference
coincides with the fact that drought severity in the latter group of sub-regions is influenced by El Niño, while in central

Sumatra and west Kalimantan, a positive IOD is the most important driver of droughts (Field et al., 2016, Pan et al., 2018).

**Figure 5: Same as Fig. 3. but for prediction of probability for active fire > 10 cases.**

ProbFire Prediction of active fire count > 10 class occurrence probability was generally less reliable and substantially less
confident (Fig. 5) when contrasted with the model reliability performance for active fire > 0 cases seen in Fig. 4. Reliability

of ERA5-based predictions for different years (Fig. 5a) exhibited much more variability. The large spread is partially





attributable to the fact that low fire activity years did not have enough active fires > 10 grid cells needed to determine reliability of probability prediction. Reliability of ERA5-based predictions for different sub-regions (Figs. 5b-f) was also slightly worse when compared to the active fires > 0 prediction diagrams. Probability estimates were noticeably underconfident for west Kalimantan (Fig. 5d) at low probability values. The biggest difference in reliability of predictions

between the two fire occurrence classes was observed for SEAS5-based issued probabilities. Reliability of high probability predictions of active fires > 10 occurrences deteriorated rapidly with lead time. Notably, low numbers of high confidence predictions limited the reliability estimation for central Sumatra and west Kalimantan sub-regions, which had very small numbers of high confidence predictions (low sharpness). This highlights that reliable and confident prediction of active fires count > 10 cases is more difficult compared to predicting active fire count > 0 cases. Low prediction confidence could be in

part attributable to low number of active fires > 10 grid cells which comprise only (~5%) of the training data set. However, the most important factor here is perhaps an intrinsic difficulty of discrimination between grid cells which do contain a few active fires (0 > active fires <11) and those in which the count exceeded 10 active fires. Fire occurrence and severity in Indonesia, besides the climatic drivers, is influenced by interplay of many location-specific factors including land management practices, policy decisions and fire suppression efforts (Page and Hooijer 2016, Tacconi 2016), none of which

could be realistically represented in a region-wide fire prediction model. Despite this difficulty, our results indicated that prediction of the active fires count > 10 category was potentially more useful for decision-makers.

**4.2 Prediction skill scores**

The model prediction metrics for active fires count > 10 cases (Fig. 6) exhibited patterns which generally followed those of the reliability diagrams. All of the scores were better for the El Niño dominated sub-regions (i.e. south Kalimantan, south

Sumatra and south Papua). By contrast, west Kalimantan and in particular central Sumatra had lower AUC, higher Brier score and substantially lower probability of detection. Importantly, not only were the climatology and SEAS5-based forecast scores worse, but also the ERA5-based predictions yielded lower validation values. This result indicates that the model was not able to optimize the classification problem as well for the latter sub-regions given the predictors used in this study. Consequently, even in the case of perfect seasonal forecasts, fire activity prediction performance would be worse in central

Sumatra and west Kalimantan when compared to the other sub-regions. Lower model skill is likely to be attributable to different dry season patterns coupled with a stronger influence of human drivers. West Kalimantan and central Sumatra in particular, do experience early season drought (in February-March) as well as the main dry season (July-September) which is common across all subregions. In contrast to El Niño dominated regions, high fire activity episodes in central Sumatra and west Kalimantan are typically shorter and do occur outside the times of the two dry seasons (Gaveau et al., 2014) (Fig. 7). As

a result, the monthly time step used by the modelling system of this study may be insufficient for resolving this rapid climatic variability.

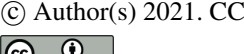
**Figure 6: Skill scores for prediction of active fires > 10 cases. Shown are mean values of area under receiver operating characteristic curve (AUC), Brier score, probability of detection and false alarm rate (figure rows) for the**
**different sub-regions (figure columns). ROC curves are shown in Figure S2 (Supplementary figures). The values for ERA5-based model predictions are shown as red bars, climatology-based predictions are depicted as blue bars, while boxplots indicate SEAS5 ensemble member prediction values at different lead times. For boxplots, shown are the median value (grey bar), interquartile range (boxplot body) and the full range (boxplot whiskers) of the SEAS5 ensemble member predictions at a given lead time.**

SEAS5-based prediction skill scores gradually degraded with increasing lead time in all sub-regions. The performance of seasonal forecasts was substantially better in the El Niño dominated sub-regions. Notably, in south Papua SEA5 ensemble predictions had both AUC and Brier scores better when compared to climatology predictions at lead times up to five months. Skill scores of SEAS5 ensemble predictions in south Kalimantan and south Sumatra indicated potential gains when compared to climatology-based model predictions at lead times of up to three months. By contrast, AUC and Brier scores of

SEAS5-based predictions in central Sumatra outperformed climatology-based model predictions only at one month lead. In west Kalimantan there was no benefit of using SEAS5 ensemble forecasts at any lead times.





**Figure 7: Difference in mean monthly Brier scores between climatology-based and SEAS5-based model predictions of active fires > 10 case occurrence predictions at lead times of 1–6 months and mean monthly active fire count for the study's subregions. Positive Brier score difference values (red shades) indicate smaller Brier values for SEAS5-based predictions (better), while negative Brier difference values (blue shades) indicate that climatology-based predictions performed better for that month and lead time. Note different colour scales for different subregions.**





These results demonstrate that ProbFire driven by SEAS5 ensemble forecasts has a relatively high potential value for the development of early warning systems in south Kalimantan, south Sumatra and south Papua. Skilful and reliable prediction of elevated fire activity five to three months in advance allows for ample time to act on the predictions. This result may be
attributable to increasingly realistic representation of ENSO-driven SST variability in seasonal forecasting models (Johnson et al., 2019).  Skill of SEAS5-based fire occurrence forecasts at one month lead in central Sumatra indicated some potential value, however, utilization of such forecasts in the early warning systems is challenging because warnings could be issued at most a few weeks before onset of a potentially elevated fire activity phase.

ProbFire predictions of monthly active fires > 10 events derived using seasonal forecasts had substantially higher probability of detection (Fig. 6) when compared to climatology-based predictions in all sub-regions. This was true for all lead times, although there was a consistent decrease in probability of detection with increasing lead time. At the same time, SEAS5-based predictions had slightly higher false alarm rates which were also increasing with lead time. Such a pattern was an expected result and is a manifestation of differences in forecasted probability sharpness. Climatology-based prediction
lacked sharpness and therefore had low probability of detection and low false alarm rates. Meanwhile, SEAS5 forecasts enabled the model to issue more confident probabilities (Fig. S1) which consequently had higher probability of detection rates but also somewhat higher false alarm rates.

### 4.3 Relative value of the forecasts

The cost-loss analysis of ProbFire fire activity forecasts demonstrated potential economic benefit of the system's fire
occurrence predictions when compared to forecasts based only on fire event climatological occurrence frequency (Figs. 8 and 9). While at least some forecasts users in all study sub-regions would have benefited to some degree, the potential maximum value and range of user cost-loss ratios that would have gained from using the system varied considerably across Indonesia. This analysis also revealed that there was a greater benefit from using forecasted probabilities of relatively rare, elevated fire activity grid cells (monthly active fire count > 10), rather than all fire-containing grid cells (active fires > 0)
(Fig. 8 vs 9).

The relative value of SEAS5-based forecasts was substantially higher than ERA5 climatology forecasts, but only for active fires > 10 case predictions (Fig. 9). By contrast, climatology-based predictions were very close to or equal in their potential economic benefits at all lead times when compared to those derived from SEAS5-based predictions for the active fires > 0
cases (Fig. 8). This was an expected result given that the number of low fire activity grid cells did not exhibit the same level of interannual variability as numbers of high fire activity grid cells did during the study period (Fig. 2). This result indicates that skilful prediction of widespread annually occurring burning can be achieved by model based on ERA5 climatology and non-climate information. However, ProbFire predictions based on seasonal hindcasts had higher potential economic benefits when predicting highly variable occurrence of elevated fire activity (grid cells with monthly active fires > 10).





Figure 8: Relative value of active fires > 0 cases prediction for users with different cost-loss ratios. Shown are relative value of ERA5-based predictions and SEAS-based (solid lines) and climatology-based (dashed lines) predictions at different lead times (columns) for different sub- regions (rows). Line shading (legend) indicates different fire prediction probability thresholds at which the relative value curves are calculated.





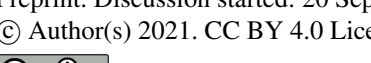

Figure 9: Same as Fig. 8. but for prediction of active fires > 10 cases.

Overall, the forecasts had the highest potential economic value for the widest range of cost-loss ratios in south Kalimantan, south Papua and south Sumatra. In these sub-regions, where the dry season severity is primarily influenced by El Niño, predictions of active fires > 10 probabilities indicate a potential gain of over 60% of the benefit from a perfect forecasting





system at cost-loss ratios close to the fire event occurrence frequency ratios (Fig. 9). SAES5-based predictions had relative values higher when compared to the climatology model at all lead times (1-6 months) and all probability thresholds. The

potential economic gain over the climatology model predictions was increasingly higher for larger cost-loss ratios. This indicates that users with larger cost-loss ratios would have benefited the most from the SEAS5-based forecasts using high probability thresholds for decision making. This was particularly true for south Kalimantan, where forecasts of active fires > 10 probabilities also indicated the highest sharpness (Fig. 4). At the same time, for cost-loss ratios over 0.5, climatology-based predictions offered little in terms of potential economic benefit.


By contrast, ProbFire forecasts for central Sumatra and west Kalimantan indicated potential benefit for narrower range of cost-loss ratios and lower total relative values of less than 60% of perfect forecast performance (Fig. 9). While SEAS5-based predictions had modestly higher relative value when compared to the climatology model, this was true only at lead time of 1 month. In addition, the potential economic benefit at 1 month lead was present for lower cost-loss ratios. This result is a

manifestation of low sharpness and reliability of SEAS5-based forecasts in these regions (Fig. 4), at longer lead times.

**5 Conclusions**

Predicting highly variable landscape fire activity is an inherently difficult problem due to the complexity of factors influencing fire dynamics at different time and space scales, and the large uncertainties associated with datasets used to characterise these fire drivers. Previous studies have shown that climate information from current state-of-art seasonal

forecasting systems can be utilized for seasonal fire prediction in parts of the globe (Turco et al., 2018), including Indonesia (Spessa et al., 2015, Shawki et al., 2017). While climate is clearly an important driver of fire activity, these climate-fire relationships are modified by human activity across a range of spatial scales, especially in regions undergoing rapid land cover changes such as Indonesia. To reflect this additional source of variability, early fire warning systems in the region need to utilise non-climate information for fire prediction.


In this study we have developed and tested ProbFire, a new probabilistic early fire warning modelling system for Indonesia which provides a blueprint for future operational warning systems in the region and beyond. Compared with previous regression-based fire forecasting studies focused on climate-fire relationships (Spessa et al., 2015, Chen et al., 2016, Turco et al., 2018, Chen et al., 2020), ProbFire integrates ECMWF ERA5 reanalysis and SEAS5 seasonal climate predictions with

non-climate features and employs multilayer perceptron classification model for probabilistic fire event prediction at 0.25 degree spatial resolution. The probabilistic approach adopted by this study is better suited for predicting rare and/or newly occurring fire events and allows the forecasts to be evaluated from a user perspective using the cost-loss model.





Validation of ProbFire performance for the 2002-2019 period showed that SEAS5-based fire event probabilities were
generally well calibrated, although as expected, the reliability of high confidence predictions gradually decreased with
increasing lead times. SEAS5-based fire predictions outperformed the climatology-based model at lead times of three to five
months in south Kalimantan, south Sumatra and south Papua, where drought severity is strongly influenced by El Niño. By
contrast, SEAS5-based forecasts for central Sumatra had higher skill scores only at one month lead times, while in west
Kalimantan they showed no improvement at all when compared to climatology-based predictions. Analysis of potential
economic benefits of the forecasts indicated that forecast users with a wide range of cost-loss ratios would have benefited
from using the SEAS5-based predictions in decision making in the El Niño dominated regions of Indonesia. This
demonstrates that early fire warning systems based on ECMWF SEAS5 seasonal climate forecasts and non-climate
information can support the work of various stakeholders involved in fire prevention and management including Indonesian
government agencies, local communities and commercial entities.


ProbFire has limitations and further research is needed to improve the skill of the predictions, especially in parts of Indonesia
that lie outside the El Niño zone of influence. Lack of predictability in central Sumatra and west Kalimantan at lead times
beyond one month indicates the generally low skill of SEAS5 climate predictions in these IOD-dominated parts of Indonesia,
and could potentially be improved by integration of seasonal forecasts from different modelling centres.


However, even the ERA5-based model had lower predictability in those areas, which indicates that different input data may
be needed. In addition, in central Sumatra severe burning episodes can be triggered by short term droughts (Gaveau et al.,
2014) which cannot be represented at the monthly temporal resolution of this system. Furthermore, the non-climate datasets
used in this study cannot represent the full range of environmental and anthropogenic factors which modulate fire occurrence
across Indonesia. Consequently, the system uses primitive identifier features for the five sub-regions as none of the used
datasets could reflect the full range of differences in fire-climate sensitivity between the sub-regions. In addition, past and
future changes in national and local policies and fire suppression efforts are currently not included in ProbFire. These
changes could affect the future performance of the system if they reduce the region's fire sensitivity to the climatic and
biogeographic features used to drive ProbFire. To address these issues, the development of long-term, consistent and
regularly-updated datasets on vegetation, land management status and socio-economic drivers of fire activity is needed.

## 6. Code availability

ProbFire source code can be found at https://github.com/ToFEWSI/ProbFire
## 7. Data availability

All datasets used in the study are publicly available. Primary forest cover in year 2000 is available at:
https://glad.umd.edu/dataset/primary-forest-cover-loss-indonesia-2000-2012. The annual forest loss data available on-line
from: http://earthenginepartners.appspot.com/science-2013-global-forest. Active fire data can be downloaded from:
https://firms.modaps.eosdis.nasa.gov/download. Peatland extent maps are available at:
http://data.globalforestwatch.org/datasets. ERA5 reanalysis and SEAS5 hindcasts can be downloaded from:
https://climate.copernicus.eu. ProbFire input datasets aggregated to 0.25 resolution can be accessed at
https://zenodo.org/record/5206278.

## Author contributions

T.N. and A.C.S designed the study. T.N. implemented the analyses and wrote the manuscript. A.C.S, S.H.D., G.D.C. and
S.M. contributed to interpreting the findings and writing the final paper.

## Competing interests

The authors declare no competing interests

## Acknowledgements

This study forms part of the Towards a Fire Early Warning System for Indonesia (ToFEWSI) project, which is funded
through the UK's National Environment Research Council – Newton Fund on behalf of UK Research & Innovation
(NE/P014801/1) and Indonesia Endowment Fund for Education and the Indonesian Science Fund.

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
