# Peer review of "ProbFire: a probabilistic fire early warning system for Indonesia"

_Natural Hazards and Earth System Sciences, 2021_

## Author Response (AR2)

The authors would like to thank the reviewers for their positive view of the manuscript and also for constructive feedback and suggestions for improvement. In addition to all the changes made in response to the reviewers comments, we have also changed the way we report lead month count of the predictions. We were previously reporting that forecast for the current month (typically issued on the 1st of the month) had lead time of 1 month. However, we have since realised that a more accepted way (and arguably more correct) is to consider predictions for the current month having lead time of 0. As a result, all lead month counts reported in the text and figures of the revised manuscript were reduced by 1 month when compared to the initial version. This change does not affect the interpretation nor implications of the study's results, but makes the way lead times are counted/reported more in agreement with the nomenclature of seasonal climate forecasting studies.

Please find below a point by point response to the reviewer comments received. Reviewer comments received are italicised while our responses are in normal font. Line numbers in our responses refer to those in the revised manuscript.

**Reviewer #1:**

*Line 179: "total monthly prediction for the five preceding months", is it necessary to use the value of the previous five months? Is the three preceding months enough considered the long-term impact of precipitation (similar to DC in FWI)? The results also show that the longer prediction time, the lower reliability.*

During the model development and testing phase we have experimented with various time windows for different climate variables. The testing results showed that it is beneficial to include precipitation for five preceding months rather than tree (which as R1 indicates here is approximately the length of precipitation "memory" of drought code). The benefit of extending precipitation record beyond three months was mainly manifested through small but tangible increase in prediction confidence for active fire count > 10 cases (improving prediction sharpness). This result was also one of the reasons why we decided against using the fire weather indices as predictors in this study in the first place (another reason being lower computational complexity). We have amended the sentence (lines 185-187) clarifying this:
"Precipitation for 5 months preceding the month of interest was included to characterise long term build-up of drought conditions, and the number of months was determined empirically during the model optimization stage."

*Line 286: Fig S3, it is recommended to adjust its order according to the cited order in text.*

Thank you for the suggestion, subplots reordered to follow the order of discussion in the Methods section.

*Line 299: "fire occurrence probabilities for 2006 were predicted and evaluated", was 2006 selected randomly?*

Here we simply used 2006 as an example to illustrate leave-one-year out dataset split strategy. The same procedure was applied to all years in the study period. We have rephrased the sentence to remove reference to a specific year: "For example, fire occurrence probabilities for year x were predicted and evaluated using data from all years except year x for model training.

*Line 300 "This resulted in 17 different realizations of the model all having different weights and biases" it'd better show the weights of the indexes in text.*

While we used a comparably simple multilayer perceptron model architecture with small number of features, the trained model nonetheless consists of 540 weights (connections between 18 x 15 x 2 neurons in the model) and 15 bias values (for each neuron in the hidden layer). As a result it is impractical to include these values in text. However, we do share the pre-trained model instances with weights and biases in the supplementary dataset (https://zenodo.org/record/5206278). The link to the dataset was added to the sentence lines (310-311), also clarifying why there are 17 model instances: "This resulted in 17 different realizations of the model (one for each year in the record) all having different weights and biases, due to different subsets of the dataset being used for training. Pretrained models with weights and biases are available at https://zenodo.org/record/5206278."

*Line 359: formula (6) is not displayed correctly, "â       2"?*

Thank you for pointing this, the formula has been corrected.

*Line 375-390: "if the event is a 'peatland fire', and the action is' fire preventive measures', then loss would equal the total economic loss", it's better to explain the main preventive measures could be used in the region? How to determine the cost/loss ratio in the model? Did you get the ratios from fire statistics data? Do you calculate by subregion and land cover type?*

The authors agree with the reviewer that including examples of fire preventative measures would be beneficial in illustrating the cost/loss model. The example of preventative measures is now added to the relevant paragraph (lines 389-394):

"In Indonesia, a range of different fire preventative actions could be utilized depending on the lead time of forecasts. Early warning (lead times of several month) would allow forecasters and relevant authorities to inform the communities in fire-prone areas, legislate to prevent agricultural fire use for the season and increase preparedness and train fire service personal. Forecasts issued at less than 1 month lead times could be utilised to implement local bans of specific fire uses (e.g. agricultural waste burning), and to deploy monitoring and fire fighting resources to the high-risk areas."

We would like to note here that one of the main properties of the model that makes it attractive as a forecast evaluation tool is the fact that the issuers of the forecast (the authors in this case) do not need to determine specific cost/loss ratios. This task would be difficult if not impossible without having access to information that only local fire management authorities have. The model simply shows potential economic value (y axis in Figs. 8 and 9) of the forecasts for a range of cost/loss ratios (x axis in Figs. 8 and 9). The task of determining the ratio is left to potential users of the forecasts; in this case policy makers and fire managers at national and local levels in Indonesia.

*Line 442ï¼ "Figure 5: Same as Fig. 3" should be as fig 4.*

Thank you for spotting this error. Corrected.

*Line 449: "for West Kalimantan (Fig. 5d)", should be Fig 5f. However, the Fig 5e shows more obvious differences, which need to be explained.*

Thank you for noting this mistake. Indeed, the reliability of predictions was low both for central Sumatra and west Kalimantan, and the figure reference was fixed accordingly. We have perhaps not made it clear enough that the interpretation on Line 449 applies only to ERA5-based predictions (red lines in Fig. 5). The sentence was rephrased. We believe that the reviewer refers to the more obvious differences in SEAS5-based predictions, which indeed were larger and more important. The interpretations of those differences is presented in the following sentences (lines 455-465)

*Line 510: Fig. S1, I suggest to put this figure in text. It is the only figure to show the predicted results in spatial.*

We agree with the suggestion. The figure was moved to the main text as figure 6.

*Line 534: "Figure 9: Same as Fig. 8. but for prediction of active fires > 10 cases", I think the figures 8 and 9 can put together and just add the corresponding legend in upper.*

We do agree that figures 8 and 9 ideally would be presented as one figure. However, we chose to separate them only because of size. While this should not matter for digital publication, but we are not convinced that they would be legible if put on one A4 page. We propose to move figure 8 to supplementary figures and only keep figure 9 in the main text.

*In conclusion: it's better to explain the specific application of the model. For example, if active fire > 10 is predicted in the next month or the next three months in a region, what specific measures can be taken in a region?*

This broader comment relates to the comment regarding lines 375-390. As discussed in our response to comment concerning lines 375-390 above, several examples of fire preventative actions which could be exercised in the region given forecasts issued at different lead times is now added to the relevant paragraph (lines 389-393)

*Some words need to be checked and modified in format, such as"km2" on line 120, "t2m" on line 193, should be "$km^2$" and "$t\_{2m}$".*

Thank you for pointing this, corrected to our best effort.

**Reviewer #2:**

*MODIS imagery is used as a basis for fire inventory. I am wondering whether this choice might introduce a bias towards large fire events, consequently leading to an underestimation of small fires? This is mentioned around line 118, but might be worth a short comment in the discussion section.*

This is a valid point as omission of small fires and consequent underestimations of fire activity is a well known problem associated with MODIS fire datasets. However, we believe that it is of little importance in the context of this study, which aims to predict elevated fire activity rather than quantify fire effects such as burned area or fire affected area. Importantly, as our results show, the number of grid cells with low monthly active fire counts (< 11) does not exhibit large interannual variability, and hence it is not of much importance for seasonal forecasting. Prediction of rare high activity fire events is the main interest of early warning systems and as a result, omission of small fires in MODIS record is less important. We have added a sentence to state this (lines 122-124):
"In any case, omission of small fires in the product is not critical for early warning systems aimed at alerting the risk of unusually high fire activity events, rather than quantify fire effects such as fire-affected area."

*l. 144: how is the threshold motivated? Is this just because 10 is a nice number, or is there evidence that counts > 10 are particularly dangerous or may indicate particularly dangerous conditions?*

The reviewer is right to point that the active fires > 10 threshold was selected based on numerical properties. However, this was done not because of number 10 per se, but because it divides the whole dataset of monthly active fire counts at 25km resolution into 95% (active fires < 11) vs 5% (active fires > 10) parts. Coincidently, monthly active fire count > 10 threshold also represents top 25% (upper quartile) of all grid cells where active fires were detected within a given month (active fires > 0). While we have no evidence suggesting that the threshold represents particularly dangerous fire conditions on the ground, but we believe it is nonetheless a useful indicator of "elevated" fire risk as it indicates the top quartile of all fire affected grid cells.

The threshold does also represent a "sweet spot" between the increasing dataset imbalance and potential usefulness of the forecasts. During the initial model testing phase we noted that at this level of dataset imbalance the model was still generating generally reliable and calibrated probabilities. While prediction of even more rare events (increasing the threshold) would have been potentially more useful, however this resulted in loss in forecast sharpness (very little to no high probability predictions) and hence reduced the potential economic value of the forecasts.

We have reworded section 2.1.1 to make the selection criteria clear (lines 129-150)

*Section 2.3: There are some indicators available through Copernicus Global Land Service (https://land.copernicus.eu/global/themes/vegetation) which might be of interest for future work. Most of them are based on Sentinel-3/OLCI or PROBA-V and are available from 2014 onwards.*

Thank you for this suggestion, the Copernicus Global Land Service is indeed a promising source of information for future fire prediction studies.

*Section 3.1: It is unclear to me whether the hyperparameters were tuned or simply set to the values reported in the manuscript? The text does not provide any indication of hyperparameter tuning (e.g. via cross validation), a quick glance into the code implies that values were simply determined beforehand. This should be stated more clearly.*

We did perform a cross validation using grid search to determine optimum number of hidden layer neurons, alpha parameter and solver for the mlp classifier. The Brier score was the benchmark for parameter selection. This is now stated in the manuscript: "The model architecture and optimal parameter setup were determined performing grid search cross-validation and evaluating the model's performance on validation data."

*Section 3.1: Minor technial nitpicking: I think the split described her refers to `train' and `test' sets, respectively. The `validation' set is usually a subset of the training dataset used for hyperparameter tuning, before testing the tuned and validated model on the unseen `test' data.*

Thank you for pointing out this discrepancy. Changed the wording accordingly.

*Section 3.6 / Formulas (3) and (4): I think it is more common to use the terms "true positive" (TP) instead of "hits"; "false negative" (FN) instead of "misses" and "false positive" (FP) instead of "falsealarms". "pod" could be "sensitivity" or "recall".*

We agree with the suggestion regarding "true positive", "false negative" and "false positive". Changed accordingly. However, the authors feel that "probability of detection" is a more intuitive term for non-specialist audiences and it makes sense to use it here, in particular given that we are dealing with rare event prediction. While "sensitivity" and "recall" are more widely used in machine learning circles, we suggest to keep "probability of detection" in the manuscript considering that the readership of the journal (geosciences, environmental sciences, policy and broader public).

*I assume that the data set is somewhat imbalanced - i.e., there are more non-event pixels than fire pixels. Was this accounted for (in terms of model formulation or in terms of performance metrics)?*

This is a very important observation and the dataset indeed is imbalanced as we discuss in Section 2.1.1. While the level of imbalance is not extreme (95% vs 5% for active fires > 10 case) it is nonetheless considerable. However, the dataset is relatively large (449280 samples overall) and the minority class is still well represented by over 21000 grid cells. We did take special care in selecting forecast evaluation metrics which are appropriate for probabilistic predictions regardless of dataset imbalance. Note that imbalance can cause problems when dealing with classification methods that turn continuous probability distribution to dichotomous predictions by setting arbitrary class cut-off thresholds and optimize models using improper scoring rules (accuracy is one example). In this study we used reliability diagrams and the Brier score (a proper score) which by definition optimizes for "true" expected event occurrence probabilities and work regardless of class imbalance (when comparing performance of different models/model parameters for the same sample of events).

We did not subject the data to any special treatment (such as oversampling the minority class) nor did we see a need to adapt the model formulation to account for imbalance. Artificial balancing of the class sizes prior to training or application of different class weights to the loss function during training would only lead to artificially inflated minority class probabilities, and consequently would have a negative impact on the performance metrics employed in this study.

*Finally, I would like to acknowledge the provision of the ProbFire source code via GitHub and the comprehensive data avilability statement.*

Thank you for the appreciation.